# The Upshot of the SARS-CoV-2 Pandemic on Nursing Assistants: Evaluating Mental Health Indicators in Huelva

**DOI:** 10.3390/jcm11092586

**Published:** 2022-05-05

**Authors:** Francisco-Javier Gago-Valiente, Emilia Moreno-Sánchez, Emilia Vélez-Moreno, María-de-los-Ángeles Merino-Godoy, Jesús Sáez-Padilla, Francisco de Paula Rodríguez-Miranda, Emília Isabel Martins Teixeira da Costa, Luis-Carlos Saenz-de-la-Torre, Adrián Segura-Camacho, María-Isabel Mendoza-Sierra

**Affiliations:** 1Nursing Department, Faculty of Nursing, University of Huelva, 21007 Huelva, Spain; francisco.gago@dstso.uhu.es; 2Department of Pedagogy, Faculty of Education, Psychology and Sports Sciences, University of Huelva, 21007 Huelva, Spain; emilia@dedu.uhu.es (E.M.-S.); francisco.paula@dedu.uhu.es (F.d.P.R.-M.); 3Medicine Department, Faculty of Medicine, University of Malaga, 29071 Malaga, Spain; eliavm3@gmail.com; 4Integrated Didactics Department, Faculty of Education, Psychology and Sports Sciences, University of Huelva, 21071 Huelva, Spain; jesus.saez@dempc.uhu.es; 5Nursing Department, Health School, University of Algarve, 8000 Faro, Portugal; eicosta@ualg.pt; 6Health Sciences Research Unit, Nursing, Nursing School of Coimbra, 3000 Coimbra, Portugal; 7Department of Social, Development and Educational Psychology, Faculty of Education, Psychology and Sports Sciences, University of Huelva, 21007 Huelva, Spain; luis.saenz@dpces.uhu.es (L.-C.S.-d.-l.-T.); adrian.segura@dpee.uhu.es (A.S.-C.); imendoza@dpsi.uhu.es (M.-I.M.-S.)

**Keywords:** burnout, nursing assistants, coronavirus infections, mental health, health workers, nursing, public health

## Abstract

Healthcare professionals who work in front-line situations are among those under the highest risk of presenting negative mental health indicators. We sought to assess the prevalence of low personal realization, emotional exhaustion, and depersonalization as well as probable non-psychotic psychiatric pathologies during the pandemic in nursing assistants in the city of Huelva (Spain), and to study the association between these mental health indicators and sociodemographic and professional variables. A cross-sectional descriptive investigation with a quantitative approach was used. A representative sample of these professionals, consisting of 29 men and 284 women, completed the GHQ-12 questionnaire, including sociodemographic data and the MBI-HSS questionnaire, collecting information on situations of contact with SARS-CoV-2. Data analysis was conducted, and correlations were established. We found that emotional exhaustion, depersonalization and probable non-psychotic, psychiatric pathologies were related to contact with SARS-CoV-2. Moreover, personal realization, depersonalization and emotional exhaustion were related to just gender. We conclude that nursing assistants from public hospitals in the city of Huelva who had contact with patients with SARS-CoV-2 in the workplace, showed poor mental health indicators than those who did not come into contact with infected individuals.

## 1. Introduction

Healthcare professionals who work in front-line care (e.g., medicine, nursing, and nursing assistants) are among those at the highest risk of presenting negative mental health indicators. This risk is approximately twice as high among health professionals when compared to other professional areas, an aspect generally related to institutional contexts and working hours, which require a more in-depth analysis for a better understanding of this phenomenon [1]. It is important to highlight that, between 2011 and 2014, the prevalence of burnout amongst healthcare workers was approximately 9%, whereas it remained stable in other professional groups [2]. It should be noted that more than 10% of all occupational disease claims are attributed to work stress [3].

This manuscript focuses on a specific professional category, nursing assistants, who perform basic functions such as providing comfort and basic care and facilitating the transfer and safety of patients from the different areas of healthcare centers. These professionals provide proximity care, which is why it is essential to know their mental health status due to the potential impact this can have on their performance, specifically on the quality and safety levels of care they can offer.

### Background

Burnout syndrome has been identified and classified by the World Health Organization as a synonym of chronic work stress [4]. Evidence associates the occurrence of burnout with several negative outcomes that are manifested in various domains of individuals’ lives, whether personally, socially, or organizationally. In healthcare, these consequences comprise aspects such as substandard attention to the patients and greater care failures [3]. This syndrome is made up of three dimensions, which constitute mental health indicators: low personal realization, high depersonalization and high emotional exhaustion [5].

These dimensions are also associated with conflictual relationships with co-workers and an intention to leave the profession. In the same vein as the consequences caused by stress, burnout experiences can also lead to physical health disorders, destabilize family dynamics, potentiate the development of addictive behaviors, and increase the risk of depression, and even suicidal ideation [6].

It is also worth highlighting the already known impact of the SARS-CoV-2 pandemic that most countries still face and the situation of health professionals working in front-line positions. Various reports concerning the outcome of this extreme event, i.e., COVID-19, on the mental health of health personnel have been published in the last 2 years [7]. However, more than the immediate results, it is important to continue to assess the medium- and long-term impact as well as the physical and emotional sequelae that this extreme event of the 21st century will leave on individuals and, particularly, on health professionals.

It is essential to analyze the consequences of this extreme event (the pandemic) on nursing assistants, since their profession is characterized by direct and close contact with sick people to improve their health, well-being, and recovery. Their technical duties include taking the temperature or managing supplies. Regarding the daily tasks, nursing assistants help patients with personal hygiene, mobility and eating when necessary, and they also provide emotional support.

In this study, we intended to identify indicators of mental health, such as self-perceived mental health, emotional exhaustion, depersonalization, and personal realization in a state of pandemic, in male and female nursing assistants who work in public healthcare centers in Huelva (Spain), and identify socio-demographic correlates of these mental health indicators.

## 2. Materials and Methods

### 2.1. Participants

The recruitment of professionals was carried out in the two public hospitals of Huelva city (Spain): Complejo Hospitalario Juan Ramón Jiménez and Hospital Infanta Elena.

This is a descriptive, cross-sectional study with a quantitative approach, with the participation of 313 nursing assistants, 29 of whom were men (9.27%) aged between 27 and 63 years, and 284 women (90.73%) aged between 22 and 64 years. The distribution by hospital was as follows: 62.94% (*n* = 197) from Complejo Hospitalario Juan Ramón Jiménez, and 37.06% (*n* = 116) from Hospital Infanta Elena. Regarding the marital status of the participants, 4.8% of the participants were divorced, 30.9% were single, 46.7% were married, 14.6% had a companion and 3% were widowed. The majority had at least one child (64.1%), and 35.9% did not have any children (Table 1).

### 2.2. Procedure

Data collection occurred from April to June 2020 and we used a probabilistic method to select the sample with an estimated error of 0.05 (99% confidence interval). We have prepared a schedule to organize several hospital visits, different shifts, and various units and services, in order to avoid repetition of data. In each appointment, 30-min questionnaires were given to every nursing assistant, in paper format. The investigator or a collaborator were present to solve any doubts. All professionals who participated were guaranteed anonymity and all of them gave their written informed consent. The response rate was 81% for the professionals of Complejo Hospitalario Juan Ramón Jiménez and 81.6% for the professionals of Hospital Infanta Elena. Those who did not respond to the questionnaire stated that they had a lot of work to do or did not have enough time to participate. We received approval from the Board of Directors of both hospitals as well as the Research Ethics Committee of the Province of Huelva (TD-EPSH-2019 and internal code 1585-N-19).

It is worth highlighting the total number of cases of SARS-CoV-2 infection reported by AIDT (Active Infection Diagnostic Tests) in the different zones of the province of Huelva, during the data collection period (Figure 1).

### 2.3. Instruments

Before participating in the study, the nursing assistants answered a set of questions aimed at sociodemographic characterization, providing information about work center, parent status, gender, marital status, and age. They were also asked for their e-mail address to prevent data duplication. At the end of the questionnaire, we included a question that aimed to assess the respondent’s possible contacts with patients with COVID, within their professional context.

Maslach Burnout Inventory (MBI) in its general version, was one of the instruments used [5], which has been internationally validated. We selected the version prepared for health personnel: MBI–Human Services Survey (MBI–HSS). Our choice of this instrument, for the recognition of mental health indicators, was motivated by its reliability indices: 0.90 for emotional exhaustion, 0.71 for personal realization, and 0.79 for depersonalization, with an internal consistency of 0.80 for all items. Likewise, factor analyses have been conducted to validate the instrument, defining a three-dimensional structure that points at the dimensions of burnout syndrome; that is, the instrument measures exactly what the study variable aims to measure [9]. This type of factor validity is supported by studies of convergent validity, which have been carried out by the same research team. Other studies that have analyzed the MBI show Cronbach’s alpha coefficients of 0.78 for the dimension of emotional exhaustion, 0.76 for personal realization, and 0.71 for depersonalization [10]. Thus, the suitability of the MBI for the current research is proven.

The quantification of symptoms in the participants was performed using a 7-point measure (0 to 6 points), for which a higher score corresponds to a higher level of symptoms. Thus, we set at ≥27 (sum of the items of this dimension) the cutoff point to identify individuals with high emotional exhaustion, and a score of ≥10 for high depersonalization. Likewise, the cutoff point for low personal realization was ≤33 points [5]. The diagnosis of a person with burnout syndrome is established when there is an affectation in all three dimensions, that is, high emotional exhaustion, high depersonalization and low personal realization. Thus, based on the three subscales or dimensions, the results are categorized in high, medium, and low levels, and the scores are indicated below:Emotional exhaustion: high (≥27), medium (19–26), and low (≤18).Depersonalization: high (≥10), medium (6–9), and low (≤5).Personal realization: high (≥40), medium (34–39), and low (≤33).

The evaluation of probable non-psychotic psychiatric pathologies (self-perceived mental health) was carried out with the General Health Questionnaire (GHQ-12). This instrument was elaborated to detect probable non-psychotic psychiatric pathologies in people [11]. The conclusions of the validation investigations in Spanish and the suggestions of the creators of the questionnaire to use a cutoff point of ≥12 in the detection of people who might have emotional or mental disorders [12,13]. The GHQ-12 has been previously validated in Spain and has been broadly used to evaluate the general population [11,12,13,14]. A different study [15] have validated the questionnaire in a pool of 1,641 people, finding an adequate internal consistency, with general Cronbach’s alpha coefficient values of 0.90. Lastly, another investigation [16], with a group of 854 people, showed that this tool has good psychometric features and reliability, with a Cronbach’s alpha of 0.80. The 12 items of the instrument also presented adequate internal consistency (Cronbach’s alpha if item deleted from item 1 to 12: 0.795; 0.788; 0.799; 0.796; 0.791; 0.776; 0.792; 0.784; 0.793; 0.776; 0.769; 0.79).

### 2.4. Data Analysis

The statistical program SPSS (Statistical Products and Service Solutions) v.23.0 was used to perform all statistical calculations (https://www.ibm.com/analytics/spss-statistics-software).

Firstly, it is important to point out that for the variables of emotional exhaustion, age, personal realization and depersonalization, a univariate analysis was carried out that included the standard deviation, the mean and the maximum and minimum value.

Secondly, the frequency and percentage were calculated for marital status, gender, parent status, emotional exhaustion, work center, depersonalization, personal realization, self-perceived general health, and contact with SARS-CoV-2.

The Kolmogorov–Smirnov statistic was also selected for normality tests. In this way, it would be determined if the tests to be passed later would be parametric or non-parametric.

Statistical tests were also carried out according to the objectives and characteristics of this investigation. The analyzes performed are enumerated below:-Cross-tabulation analyses for depersonalization, emotional exhaustion, and personal realization as a function of gender, work center, contact with SARS-CoV-2, marital status, and parent status. Chi-squared tests were also performed between these variables.-Since the normality tests for depersonalization, age, personal realization, and emotional exhaustion showed normal distribution, the Mann–Whitney U-test was chosen for the independence tests with categorical variables of two categories, and the Kruskal–Wallis H-test for three or more groups. We also examined the correlations between the different defined variables, through Spearman’s Rho.-Cross-tabulation analyzes based on the results obtained in the GHQ-12 due to depersonalization, emotional exhaustion, contact with SARS-CoV-2 and personal realization. Chi-square tests were also performed between these variables.-Cross-tabulation analyses based on the results obtained in the GHQ-12 as a function of sex, marital status, parent status and age. Chi-square tests were also performed between these variables.

## 3. Results

### 3.1. Relationship of Emotional Exhaustion, Depersonalization, and Personal Realization with Gender

The *p*-value of the Chi-squared independence test was significant (χ^2^ = 10.320 and *p* = 0.006 for emotional exhaustion; χ^2^ = 58.728 and *p* = 0.000 for depersonalization; χ^2^ = 18.419 and *p* = 0.000 for low personal realization). Therefore, with 95% confidence level, we can accept the dependence hypothesis between the variables, that is, the affectation in the different dimensions varies according to gender. Larger percentages of high depersonalization, high emotional exhaustion, and low personal realization were presented by men when compared to women (Figure 2).

### 3.2. Prevalence of Depersonalization, Emotional Exhaustion, and Personal Realization as a Function of Work Center

There was a larger percentage of affectation in Complejo Hospitalario Juan Ramón Jiménez (44.6%) with respect to Hospital Infanta Elena (38.4%) in emotional exhaustion. In depersonalization and low personal realization, Hospital Infanta Elena presented greater percentages of affectation (35% and 29.4%, respectively) than Complejo Hospitalario Juan Ramón Jiménez (27.4% and 22.3%, respectively).

### 3.3. Correlation of Emotional Exhaustion, Depersonalization and Personal Realization with Age, Marital Status, and Parent Status

In the Kruskal–Wallis independence tests, the *p*-value was significant for age with emotional exhaustion and depersonalization (Kruskal–Wallis H = 28.557 for emotional exhaustion and 22.959 for depersonalization; *p* = 0.000 at 95% confidence level in both dimensions). Thus, there are statistically significant differences between the affectation of emotional exhaustion and depersonalization with respect to age. However, when the correlation coefficient between these dimensions with age was analyzed, a positive correlation was shown, although this correlation was very weak (Table 2).

Regarding the marital status, it was observed that the *p*-value of the Chi-squared independence test was significant in relation to depersonalization, emotional exhaustion, and personal realization (χ^2^ = 67.688 for emotional exhaustion; χ^2^ = 60.043 for depersonalization; χ^2^ = 76.535 for personal realization; *p* = 0.000 for the three dimensions, with 95% confidence level). The participants who showed the highest percentage of affectation in emotional exhaustion were those who had a partner (53.7%), whereas the ones who presented the lowest percentage were those who were widowed (0%). The participants with the highest percentage of affectation in depersonalization were married (33.7%), and those with the lowest percentage were divorced (23.5%). Lastly, the highest percentage of affectation in low personal realization was obtained by the group of participants who were widowed (71.4%), and the lowest percentage belonged to the group of divorced people (19.6%).

The *p*-value of the Chi-squared independence test was not significant for parent status with respect to emotional exhaustion, depersonalization, and personal realization ((χ^2^ = 4.360, *p* = 0.113), (χ^2^ = 4.154, *p* = 0.125), and (χ^2^ = 2.373, *p* = 0.305), respectively, with 95% confidence level).

### 3.4. Correlation of Emotional Exhaustion, Depersonalization and Personal Realization with GHQ-12 Results and Situations of Contact with SARS-CoV-2 in the Workplace

Participants who came into contact with individuals infected with SARS-CoV-2 had higher levels of emotional exhaustion and depersonalization (49.6% and 38.3%, respectively) compared to those who never had such situations (34.3% and 21.1%, respectively). However, the participants who never had contact with situations of SARS-CoV-2 presented a higher percentage of affectation in low personal realization (51.5%) compared to those who did have contact (40.5%). The Chi-squared test showed a relationship between these variables (χ^2^ = 51.059 in emotional exhaustion; χ^2^ = 34.273 in depersonalization; χ^2^ = 26.298 in personal realization; *p* = 0.000 in the three dimensions with 95% confidence level).

The participants with positive GHQ-12 also obtained greater percentages of affectation in emotional exhaustion and depersonalization than those with negative GHQ-12. However, professionals with a negative GHQ-12 result presented lower percentages of high personal realization than those with a positive GHQ-12. The variables were correlated in the Chi-squared test, with 95% confidence level (Table 3).

### 3.5. Correlation of Probable Non-Psychotic Psychiatric Pathology (Positive GHQ-12) with Gender, Marital Status, Age, and Parent Status

There was a lower percentage of women with positive GHQ-12 results compared to men. However, with 95% confidence level, no relationship was observed between these two variables in the Chi-square test (Table 4).

The greatest percentage in positive GHQ-12 was represented by windowed participants, whereas the participants who had a partner presented the lowest percentage. A relationship was observed in the Chi-squared test between these two variables (Table 4).

Regarding the GHQ-12 result as a function of parent status, it was observed that the participants without children represented a greater percentage in the positive GHQ-12. However, there was no relationship between these two variables (Table 4).

Significant results (*p* = 0.041 with 95% confidence level) were obtained in the Mann-Whitney U-test (115,915.500) when analyzing the existence of differences between the GHQ-12 data and age. Those who scored positively on the GHQ-12 had a mean age of 45.4 years, which was lower than those who scored negatively for the same variable (46.7 years).

### 3.6. Correlation of Probable Non-Psychotic Psychiatric Pathology (Positive GHQ-12) with Situations of Contact with SARS-CoV-2 in the Work

Data evidenced a higher percentage of participants with positive GHQ-12 who had been in a situation of contact with SARS-CoV-2 with respect to GHQ-12 positive participants who had never been in such situations (Figure 3). In addition, the *p*-value of the Chi-squared independence test was significant (χ^2^ = 62.483 and *p* = 0.000, with 95% confidence level), representing that these two variables are dependent. That is, the existence of possible non-psychotic psychiatric cases varies according to contact with SARS-CoV-2.

## 4. Discussion

The conclusions of preceding investigations [17,18,19] are consistent with most of the results achieved in the present research. It is worth highlighting the significant gender differences identified in relation to emotional exhaustion and depersonalization, indicating that women experience lower levels in these variables. Some studies [20] attribute high emotional exhaustion to low personal realization, with men being the most affected. These variances could be grounded on the connection of gender to other sociodemographic variables. Being male or female may imply a greater probability of the existence of other variables that would act as mediators of depersonalization and emotional exhaustion [21].

Unlike in other studies [22], no relationship was detected in the present investigation between gender and GHQ-12 results. However, this mental health indicator follows the tendency of other indicators analyzed in the study, in which men show greater affectation than women.

The greater percentage in high emotional exhaustion among the participants of Complejo Hospitalario Juan Ramón Jiménez could be due to the fact that, even though both hospitals provide the identical facilities, the palliative care service, and the unit of assisted chronic patients of Complejo Hospitalario Juan Ramón Jiménez has capacity for more patients and professionals. Thus, in line with what has been stated by other authors [19], we can conclude that these professionals suffer greater emotional exhaustion due to the amount of care they provide, which involves contact with people suffering from pain or who are dying. Other studies [23] have related low personal realization to depersonalization. This was notoriously more frequent among participants from in Hospital Infanta Elena, where most professionals presented affectation in emotional exhaustion.

There are also professionals who show high personal realization, despite having high depersonalization, high emotional exhaustion, and a poor self-perception of general health; it seems that the possibility of achieving goals and feeling useful in their work helps to overcome the challenges that there arise [24]. This fact can make these symptoms go unnoticed by the people who suffer from them.

There are contradictory results in the literature in relation to age with depersonalization or emotional exhaustion. The recognized relationship between age and these variables has been clear in some studies [25], and not so much in others [26]. However, it is important to take account that conclusions regarding age must always be considered carefully, due to the problem of survival bias; that is, those who present burnout at the beginning of their careers can leave their jobs, leaving behind those who endure and persist, who thus present lower levels of burnout [27].

Studies conducted in populations with occupations dedicated to the care of other people have concluded that the health perception of people improves with the age of the professional; however, such studies have not found statistically significant differences [19]. This tendency was also observed in the sample of the present research.

Regarding marital status, some studies [28] show higher levels of depersonalization or emotional exhaustion in participants with a partner or spouse, whereas other studies report higher levels in single participants [27]. In view of the results obtained in this study, we consider that studying the marital status alone as the single influencing factor of the family on the work life could involve a series of biases. Family support does not inevitably have to come from the partner, since people can obtain support from the parents, nephews, nieces, cousins, and other significant persons. Other studies on healthcare professionals have reported concordant results with those reached in the present study, even detailing fewer mental health problems in people who had a partner compared to those who did not have a partner [27].

The results obtained in the GHQ-12 as a function of parent status and marital status indicate that the family setting could be behaving as a controlling variable, also in the psychic health of the nursing assistants, which would explain why these symptoms went unnoticed in the people who presented these affectations [21,29].

With respect to parent status, although children could be expected to have a negative impact on the health personnel of their parents, and thus on the personal realization of the latter due to a lack of time [30], the study population did not show any results in this regard. It would also be important to analyze how the participants evaluate the family reconciliation plans received in each hospital. 

The correlation identified between positive GHQ-12 and depersonalization provides pertinent data that must be taken into account. Depersonalization can be a sign of major depression. It has been reported that, in people suffering from unipolar depression, the signs of depersonalization that are present are more exacerbated than in individuals without underlying mental pathology, and that there is a positive correlation between depression and depersonalization [5]. These problems may be related, they may even have common biological underpinning, and may be at least part of a spectrum of affective disorders [31].

Currently, there is already some research on the influence of the pandemic on the mental well-being of professional career; however, as it is a still active situation, in which new variants of SARS-CoV-2 continue to appear, we consider it important the conclusions of this investigation. It therefore appears that, among the studies that have already explored this topic, there is detailed information in consonance with the data obtained in the present research. The data reveals high percentages of healthcare professionals who have provided care or been in contact with COVID patients and who have symptoms of anxiety, stress, depression, and sleep disturbances [32,33,34].

Even before the pandemic, emotional exhaustion and depersonalization were found to be more common in men than in women. [35]. Nevertheless, the incidence of these two problems in the population studied in this investigation was higher [36,37].

### 4.1. Implications for Practice

The results found, highlight the importance of implementing mitigation policies in pandemic states. Considering that resources can be especially sparse during these extreme events, timely mental support can be made available through various strategies, including telemedicine and support groups [7].

Nursing assistants has basic functions in health care, such as providing comfort and basic care, facilitating the transfer and safety of sick people in the nursing, hospital and/or outpatient areas. Therefore, it is essential that these people maintain a good state of mental health to provide quality care.

In addition, the data highlight the need to consider the gender perspective as a path to a paradigm transformation, which allows for the development of broader and more inclusive scientific knowledge, as well as a more egalitarian, fair, and sensitive health system.

### 4.2. Limitations

Notwithstanding the importance and impact of the obtained data, we can also point some limitations. Given that it was a cross-sectional study, as it measured the relationship among the selected variables at a concrete time frame and no follow-up was carried out, many of the evaluated aspects may have suffered modifications, as is shown by the changes that are taking place in the Spanish public healthcare system [38], which may change many objective and subjective indicators of health. Furthermore, we also considered it timely to explore the probable antecedents of mental health of the participants and include questions about access to resources (economic, health, social) in this type of extreme events.

This pandemic had an important impact on individuals belonging to low socioeconomic strata. [39]. Thus, it would be relevant to carry out prospective investigations, with similar study sample, expanding the variables studied, reassessing the situation and the impact of changes implemented in the meantime. Another limitation is related to the sample magnitude. Although the number of health professionals did represent the reality of this occupational group (nursing assistants) in the hospital workers, caution is needed in extrapolating the results to other contexts (hospitals and regions).

To sum up, it is crucial to carry out new studies with a larger number of participants, to ensure results external validity. It would also be timely to analyze the correlation of the GHQ-12 score with personal realization, depersonalization, and emotional exhaustion. It would also be interesting to carry out studies of subgroups within the work centers.

## 5. Conclusions

In view of the results, we can then point out that nursing assistants of Huelva’ public hospitals who had been in contact with COVID patients in their work setting present a worse mental health level than professionals who did not have to provide care to patients with this pathology. This state is evidenced through positive result in probable non-psychotic psychiatric pathologies, high depersonalization, and high emotional exhaustion. There is also a relationship between these variables.

Secondly, it should be noted that the men of the study sample presented, in general, a worse mental health state than women.

## 6. Relevance Statement

As a strength of this study, very novel information was obtained about mental health indicators in a population of healthcare professionals, who have been poorly studied in a pandemic situation to date—nursing assistants. Therefore, this study updates the data on this topic to subsidize to future interventions in this context. This paves the road to broader investigations in the same line.

Another relevant strength of this investigation is that it evidences the unequal affectation in the mental health state as a function of gender, thereby corroborating the importance of take account the gender perspective in health science research.

Therefore, we highlight the urgent need to enable results to be used with an educational and psychosocial approach, to change vulnerable targets through the development and application of training and preventive programs.

## Figures and Tables

**Figure 1 jcm-11-02586-f001:**
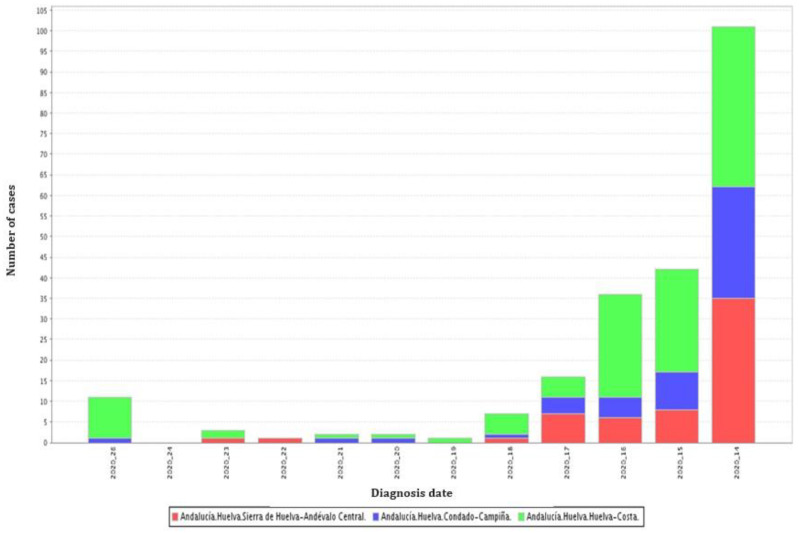
Cases of COVID-19 in the zones of the province of Huelva grouped by weeks [8].

**Figure 2 jcm-11-02586-f002:**
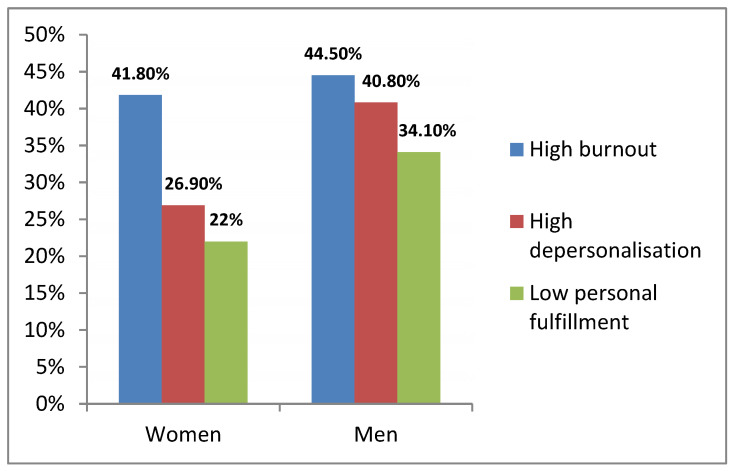
Percentages of high depersonalization, high emotional exhaustion, and low personal realization in men and women.

**Figure 3 jcm-11-02586-f003:**
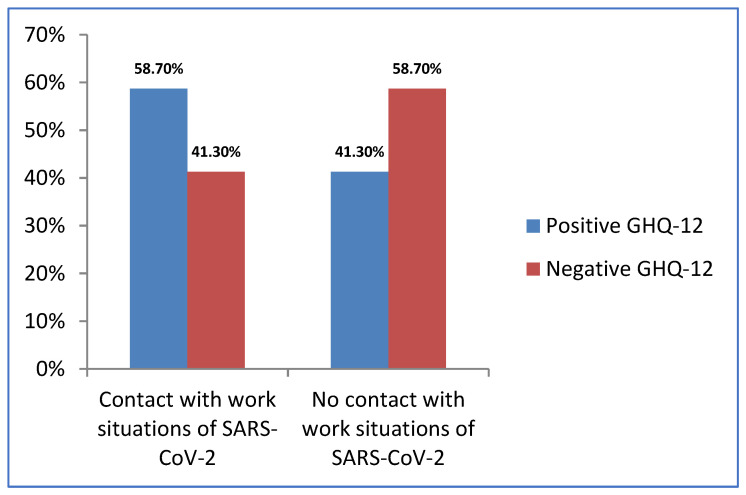
Results of GHQ-12 as a function of the situations of contact with SARS-CoV-2.

**Table 1 jcm-11-02586-t001:** Sociodemographic variables.

Sociodemographic Variables					
Sex	Men (%)			Women (%)	
	9.27%			90.73%	
Age range	Between 27 and 63 years			Between 22 and 64 years	
$\bar{x}$	51.1959			44.4835	
Work center	Complejo Hospitalario Juan Ramón Jiménez (%)			Hospital Infanta Elena (%)	
	62.94%			37.06%	
Marital status	Married (%)	Single (%)	Divorced (%)	Widowed (%)	Companion (%)
	46.7%	30.9%	4.8%	3%	14.6%
Parenthood	Yes (%)			No (%)	
	64.1%			35.9%	

**Table 2 jcm-11-02586-t002:** Spearman’s Rho correlation for the variables of depersonalization and emotional exhaustion with age.

Spearman’s Rho		Age	Emotional Exhaustion	Depersonalization
Age	Correlation coefficient	1.000	0.157	0.035
	Sig. (bilateral)		0.000 **	0.266
	*n*	313	313	313
Emotional exhaustion	Correlation coefficient	0.157	1.000	0.363
	Sig. (bilateral)	0.000 **		0.000 **
	*n*	313	313	313
Depersonalization	Correlation coefficient	0.035	0.363	1.000
	Sig. (bilateral)	0.266	0.000 **	
	*n*	313	313	313

** The correlation is significant at 0.01 (bilateral).

**Table 3 jcm-11-02586-t003:** Group statistics and Pearson’s Chi-squared test for the variables of emotional exhaustion, depersonalization, and personal realization ^a^.

					Possible Non-Psychotic Psychiatric Case
		Yes	No	Pearson’s Chi-Squared	Asymptotic Significance (Bilateral)
		%	%		
Emotional exhaustion	High	58.7%	28.5%	167.362	0.000 *
	Medium	33.0%	28.7%		
	Low	8.4%	42.8%		
Depersonalization	High	37.2%	25.0%	30.645	0.000 *
	Medium	18.8%	32.7%		
	Low	44.1%	42.3%		
Personal realization	High	31.5%	19.5%	21.725	0.000 *
	Medium	27.3%	36.9%		
	Low	41.1%	43.6%		

^a^ Grouping variable: GHQ-12; * *p*-value of the Chi-squared test.

**Table 4 jcm-11-02586-t004:** Group statistics and Pearson’s Chi-squared test for the variables of marital status, sex, and parent status ^a^.

		**Possible Non-Psychotic Psychiatric Case (GHQ-12)**			
**Independent Variables**		Yes (%)	No (%)	Pearson’s Chi-Squared	Asymptotic Significance (Bilateral)
Sex	Men	48.6%	51.4%	0.076	0.782 *
	Women	47.6%	52.4%		
Marital status	Married	52.6%	47.4%	23.588	0.000 *
	Single	48.1%	51.9%		
	Divorced	39.2%	60.8%		
	Widowed	64.3%	35.7%		
	With a partner	32.0%	68.0%		
Parent status	Yes	46.2%	53.8%	1.909	0.167 *
	No	50.7%	49.3%		

^a^ Grouping variables: results of GHQ-12; * *p*-value of the Chi-squared test.

## Data Availability

Further data that support the findings of this study are available upon reasonable request from the corresponding author. Some data are not publicly available due to privacy or ethical restrictions.

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
