# Peer review of "The Upshot of the SARS-CoV-2 Pandemic on Nursing Assistants: Evaluating Mental Health Indicators in Huelva"

_jcm, 2022, doi:10.3390/jcm11092586_

Round 1
Reviewer 1 Report
This valuable study focuses on the current state of mental health of healthcare workers, especially nursing assistants, during the global pandemic of COVID-19. In particular, the impact of contact with SARS-CoV-2 on the mental health of medical assistants is assessed using the MBI-HSS and GHQ-12 questionnaires. Although this study is a cross-sectional study and focused on a limited geographic area, it is commendable that it provides one indicator of the direction of future approaches regarding mental health aspects for medical assistants during a pandemic.
Major Comments
The research design of this study is based on a quantitative approach: 313 nursing assistants have been surveyed. The survey items include an attribute survey, the MBI-HSS questionnaire, and the GHQ-12 questionnaire. However, within this paper, there is no clear presentation of the most fundamental research data of the study. At the very least, the attributes of the 313 respondents and their respective survey data should be presented in tabular form. The presentation of data is essential, as it is a sign of the study's credibility.
The subjects analyzed in this study belong to "1" Hospital and "2" Hospital; the characteristics of their affiliation with the two hospitals are discussed in the Discussion part (L 318-326). Although differences in affiliation ("1" Hospital vs. "2" Hospital) are considered essential, statistical analysis methods such as subgroups were not used in this study. Ⅰthink that the different affiliations of the subjects should be considered, but this has not been addressed in this study. Statistical analysis methods should be considered again.
The total number of subjects analyzed in this study was 313. The process by which the 313 respondents were analyzed should be clearly presented. In general, figures such as survey collection or dropout rates should be indicated. If possible, the flow of the study should be shown using a flowchart of the study design to facilitate understanding of the flow of the study.
Reviewer 2 Report
See attached. My comments can be seen in the comments section of the PDF.

Round 2
Reviewer 2 Report
Dear authors
Thank you for comprehensively attending to my comments.
Please note two minor matters:
- I could not find the mean and SD for age
- I am puzzled by the Cronbach's alphas reported for individual items of the GHQ-12. Alpha is a measure of the internal consistency of a SCALE not individual items. Are you perhaps referring to "Alpha if item deleted"
Good luck with your research
